# Subwavelength Imaging in Sub-THz Range Using Dielectric Waveguide

**DOI:** 10.3390/s25020336

**Published:** 2025-01-09

**Authors:** Paweł Komorowski, Przemysław Zagrajek, Mateusz Kaluza, Andrzej Kołodziejczyk, Sławomir Ertman, Adrianna Nieradka, Mateusz Surma, Agnieszka Siemion

**Affiliations:** 1Institute of Optoelectronics, Military University of Technology, gen. S. Kaliskiego 2, 00-908 Warsaw, Poland; przemyslaw.zagrajek@wat.edu.pl; 2Faculty of Physics, Warsaw University of Technology, Koszykowa 75, 00-662 Warsaw, Poland; mateusz.kaluza.dokt@pw.edu.pl (M.K.); andrzej.kolodziejczyk@pw.edu.pl (A.K.); slawomir.ertman@pw.edu.pl (S.E.); adrianna.nieradka.dokt@pw.edu.pl (A.N.); mateusz.surma.dokt@pw.edu.pl (M.S.); agnieszka.siemion@pw.edu.pl (A.S.)

**Keywords:** THz radiation, self-imaging, Talbot effect, dielectric waveguides

## Abstract

Terahertz radiation patterns can be registered using various detectors; however, in most cases, the scanning resolution is limited. Thus, we propose an alternative method for the detailed scanning of terahertz light field distributions after passing simple and complex structures. Our method relies on using a dielectric waveguide to achieve better sampling resolution. The optical properties of many materials were analyzed using time-domain spectroscopy. A cyclic olefin copolymer (COC) was chosen as one of the most transparent. This study contains a characterization of the losses introduced by the waveguide and a discussion of the setup’s geometry. As a structure introducing the radiation pattern, a 2D quasi-periodic amplitude grating was chosen to observe the Talbot effect (self-imaging). Moreover, some interesting physical phenomena were observed and discussed due to the possibility of detailed scanning, with subwavelength resolution, registering the terahertz wavefront changes behind the structure.

## 1. Introduction

Significant progress in the development of sources and detectors of terahertz radiation has been made over the last few decades. Researchers have found applications in many areas such as radio astronomy [1], spectroscopy [2], security systems [3], and non-destructive inspection [4,5,6]. At the same time, various beam-shaping structures have been proposed and experimentally verified [7,8,9]. The elements that have been investigated, such as diffractive elements, allow for effective radiation propagation in a free space. When the THz wave is planned to be delivered from point to point, waveguides are often used. A solution typical for microwaves—metallic waveguides—is implemented for lower frequencies. A similar realization is well known for visible light and infrared, i.e., optical fibers made of dielectric materials. For almost twenty years, researchers have tried to apply dielectric materials to guiding a terahertz wave [10,11]. Various structural realizations (e.g., bulk and porous materials) [12] as well as waveguide shapes (e.g., rectangular and cylindrical) [13,14] were proposed. Solutions with a very efficient but rigid dielectric waveguide (DWG) were developed [15] in contrast to the elastic polymer-based realization, which is easier and cheaper to fabricate. Their characterization shows a low attenuation and a wide frequency band [16,17], which, in combination with low costs, creates many possible applications. For high-speed communication purposes, dielectric waveguides can be used as interconnection between separate circuit boards [18,19,20,21] as well as for chip connection [15,22,23]. For non-destructive inspection, they are applied in combination with a Frequency-Modulated Continuous Wave (FMCW) radar head [24] to obtain enhanced in-plane resolutions. The imaging implementation of DWG seems to be very promising. Waveguide scanning systems have also been successfully used for medical imaging [25,26,27] to illuminate the sample, as well as working in a reflection configuration. Recent studies have focused on wavefront manipulation to enable the generation of complex radiation distribution patterns. Advances in this area include the use of multi-focal metalenses [28], structures designed through iterative algorithms [29] and neural networks [30,31], various types of diffractive optical elements [8,32], as well as the exploration of other innovative methods.

Considering many applications and the necessity of proper beam-shaping, there is a great need to probe the space distribution of the THz beam formed by various optical structures. In many situations, complicated patterns cannot be visualized with a matrix of detectors due to its inappropriate pixel spacing or the correlated detector response. Raster scanning is a possible way to resolve this disadvantage, but a single detector, even very small, placed in a holder can affect the e-m field distribution. In such a situation, integration of the detector with a DWG and application of the free end of the DWG as a probing tip could solve the problem. This realization was investigated and is described in this paper.

In this study, two registering approaches were used to scan the complicated intensity pattern formed after the periodic element using a THz camera (32 × 32 pixels) and a waveguide scanning method. The latter used the waveguide to redirect the radiation to the detector placed outside the signal propagating along the optical axis, which allowed it to not form additional interferences. A detector with a larger dynamic range was used to improve the quality of imaging, and the waveguide enabled registering the complicated wavefront (Talbot effect, also called self-imaging) with high resolution. The waveguide scanning method enabled the comparison of the registered radiation with theoretical predictions and the observation of details formed in the intensity pattern. To the best of our knowledge, the optical Talbot effect has not yet been presented for such a long wavelength (3.15 mm, which corresponds to the frequency of 95.3 GHz). Alongside this research and yet independently of it, a preprint has been published showing the Talbot effect for the frequency of 331 GHz [33]. This indicates a growing interest in demonstrating and implementing various effects known from optics in the THz spectral range, driven by multiple possible applications in imaging, scanning, or telecommunication. Also, in this work, the imaging method is innovative. We describe the resolution improvement obtained with the waveguide scanning method instead of a typical matrix camera. The unconventional waveguide (rather than a metallic one) was applied for this purpose to disturb the electromagnetic field as little as possible. Combining the small waveguide size with oversampling allows us to obtain a much better image resolution.

In Section 2, we describe the dielectric waveguide used in experimental verification. Parameters of such waveguides, as well as the sub-THz transmission capabilities, are presented in the same section. The rest of this paper is structured as follows: in Section 3, we show the objects used for self-imaging; then, in Section 4, we present simulations of the Talbot effect, followed by a description of the experiment in Section 5, experimental results in Section 6, and our conclusions in Section 7.

## 2. Waveguides

Part of this research aimed to verify the possibility of using polymer filaments known from FDM (fused deposition modeling) 3D printing for transmitting radiation in the THz and sub-THz ranges. The research began with analyzing the available materials in terms of optical parameters in the considered range of electromagnetic radiation. The terahertz time-domain spectroscopy (THz-TDS) technique was used for this purpose. On the basis of the results obtained, filaments with the most promising parameters were selected for further research. The possibility of transmitting sub-THz radiation using the dielectric waveguide mechanism was demonstrated for selected filaments. Then, several experiments were carried out to describe the parameters of this transmission, the most important of which were the divergence of the beam leaving the waveguide (numerical aperture) and waveguide attenuation. Below, the results of individual tests will be discussed in greater detail.

### 2.1. Potential Filaments

Pill-shaped samples were 3D-printed with FDM technology from polymer materials in a filament form. Using the THz-TDS method, we measured and determined the optical properties of the samples in the THz spectral range. The results obtained for selected materials with various optical properties are presented in Figure 1. The data presented concern the refractive index and the absorption coefficient in the frequency domain (ranging from 100 GHz to 1 THz) of acrylonitrile butadiene styrene (ABS), cyclic olefin copolymer (COC), high-impact polystyrene (HIPS), polylactic acid (PLA), polypropylene (PP), and polyvinyl alcohol (PVA) materials. The optical properties of other polymer and composite materials are described more precisely in our previous study [34].

In this study, the crucial parameter of the materials is the absorption coefficient, which determines the maximum length of the waveguide at which the radiation will not be significantly attenuated. It should be underlined that the absorption coefficient of various materials increases significantly with frequency. However, PP, HIPS, and COC materials possess relatively low absorption coefficients throughout the analyzed spectrum, ranging from 100 GHz to 1 THz. Taking into account the low absorption coefficient of the mentioned group for lower frequencies, PP, HIPS, and COC are desired for the application described in this study. Moreover, their low absorption coefficient for higher frequencies (especially for COC material) enables their application for higher-frequency regions. Furthermore, all three materials are characterized by a refractive index in the range of 1.45 to 1.55, which is close to the values known from glass optical fibers operating in the visible and infrared radiation ranges. Therefore, the COC filament was chosen as the most promising candidate for the waveguide. The diameter of the selected filament was equal to 1.75 mm. The area roughness parameters were used to define the nonuniformity of the waveguide surface. According to the manufacturer’s description of the material, the arithmetical mean height is $S_{a}=0.25\;\mu$m, and the maximum height is $S_{z}=15.21\;\mu$m. These values are much smaller than the wavelength of transmitted radiation (3.15 mm). Therefore, one can expect that they do not significantly contribute to transmission losses.

The measurements were also conducted on the composite material—carbon fiber-filled polyethylene terephthalate glycol (PETG)-based polymer blend. Carbon fiber additives significantly increase the absorption coefficient of the polymer base. Thus, the material is an excellent absorber that does not transmit terahertz radiation. This behavior gives the opportunity for application in attenuating object fabrication, such as radiation-shielding objects [35]. However, the material was used in this study to manufacture binary amplitude masks, as precisely described in Section 3.

### 2.2. Numerical Aperture

The numerical aperture of the filament waveguide was examined using the Tera-1024 camera from Terasense Group, Inc., San Jose, CA, USA.. Radiation at a frequency of 95.3 GHz (emitted from an impact ionization avalanche transit-time (IMPATT) diode-based source) was introduced directly into the waveguide. To obtain an efficient coupling, a dedicated connector was 3D-printed. It consisted of a cone made of PP that mimics the internal shape of an antenna. The cone has a cylindrical hole along its axis that the COC filament fits exactly into. PP is a unique material that is relatively transparent and has a refractive index lower than COC. In this way, the waveguiding effect is enforced within the coupler, which improves its efficiency. Behind the cut end of the filament, a camera was placed on a motorized stage, allowing it to be automatically moved along the waveguide axis (Figure 2b). As shown later, a significant part of the THz optical power propagates outside of the waveguide. Therefore, the transmission is very sensitive to any objects close to it. The proper holding of the waveguide was already challenging, but it was achieved using low-density foams in the direct surroundings of the waveguide.

The radiation distribution at various distances behind the waveguide was recorded using a camera (Figure 2a). In the postprocessing step, the obtained images were binarized, and the number of bright pixels was calculated to estimate the area of the disk and, subsequently, its radius. The automatic threshold binarization was applied to mitigate the influence of the overexposed pixels in the scans near the waveguide. On the other hand, overexposing pixels in the nearest scans allowed for the registration of full beam width in the farther scans without changing the exposure parameter. The dependence of the radius of the observed (approximately circular) spot on the distance determines the angle of divergence of the radiation leaving the waveguide. In order to improve the accuracy of the measurement, the rays of the recorded beam were read at several dozen distances behind the waveguide. These results were plotted on a single graph and then a straight line was fitted to them. The slope of this line determines the tangent of the half-angle of divergence, as can be seen in Figure 2. The obtained half-angle is *θ* = 16.6^∘^, which translates into the numerical aperture of $NA=n_{air}\cdot sin\left(\theta \right)\approx 0.29$. The obtained half-angle value can also be interpreted as half the acceptance angle of the waveguide scanning method.

### 2.3. Attenuation

Waveguide attenuation was measured by examining the radiation intensity leaving the waveguide depending on its length. For this purpose, the end of a longer section of the tested filament was successively cut off, and the signal strength was measured. The scheme of the setup for attenuation measurements is shown in Figure 3a, while the obtained data for the COC filament are shown in Figure 3b.

The setup consisted of a monochromatic radiation source, emitting at 95.3 GHz, two 3D-printed waveguide couplers, waveguide-under-test, and a detector.

The following expression describing the attenuated power in the dielectric was fit to the experimental points:(1)$$I=I_{0}\cdot exp\left(-\alpha x\right),$$
where *I* is the measured power, $I_{0}$ is the input power, α is the waveguide absorption coefficient, and *x* is the length of the waveguide section. As shown in Figure 3, the uncertainty of the measured power is substantial. It is mostly related to the manual cutting of the waveguide. Nevertheless, through acquiring an appropriate number of experimental points, one can fit the attenuation curve and approximate the losses within the waveguide. The determined absorption coefficient is $\alpha =0.008\;{\mathrm{cm}}^{-1}$. The TDS measurements of the 3D-printed COC pills (Figure 1) show that the absorption at 100 GHz equals to $0.09\;{\mathrm{cm}}^{-1}$. However, one must note that the sample measured in TDS was 3D-printed and therefore had a structure different from pure filament. Moreover, the frequency of 100 GHz is the edge of the TDS scanning range, and the results obtained may be distorted. The extrapolation of the data obtained from the central part of the scanning range indicates a value closer to $\alpha_{COC}=0.04\;{\mathrm{cm}}^{-1}$, which should be even further reduced due to the presence of Fresnel reflections on the surfaces of the measured pill and the layers of the 3D-printed material. On the other hand, the effective absorption experienced by the radiation within the waveguide is also smaller than the absorption coefficient of the material because some part of the radiation propagates outside the waveguide, according to the transverse profile of a particular mode. In order to investigate this aspect more closely, the numerical simulations of the mode profile were performed. The full vector finite element method was utilized, provided by the Comsol 6.2 software package. The following waveguide parameters were assumed: a refractive index of the core, $n_{core}=1.52$; refractive index of the cladding (air), $n_{clad}=1.00$; core radius, $r_{core}=0.875$ mm; and wavelength, λ=3.146 mm. Such parameters resulted in a normalized frequency V=2.00, which indicates a single-mode operation. The results of the simulations are presented in Figure 4.

The data obtained (Figure 4a) suggest the non-circular mode profile with two symmetrical bulbs on the waveguide surface along the polarization direction. In such a solution, only 28% of the normalized electrical field is confined within the waveguide. It means that the effective absorption coefficient for the propagating mode approximately equals $\alpha_{eff}=\frac{1}{3}\alpha_{COC}$. The calculated $1/e^{2}$ diameters of the mode for both polarizations are equal to 3.44 mm and 4.09 mm, respectively. The dependence of the mode profile on the radiation frequency is shown in the insets (Figure 4b), together with the effective refractive index and the attenuation coefficient. For the purpose of these calculations, the attenuation of air at 95 GHz has been assumed to be equal to $\alpha_{air}=1$ dB/km, according to Protat et al. [36]. This value is most likely excessive, as laboratory air quality is much higher than cited atmospheric air with ice particles. Nevertheless, it is still a few orders of magnitude lower than the attenuation of the COC, and it is not crucial in the calculations. The obtained mode attenuation coefficient is equal to 2.62 dB/km at 3.146 mm wavelength.

Taking into account all of the above considerations, one can conclude that the value of the effective absorption coefficient obtained through the cutting method is by all means reasonable. This proves that the tested filament exhibits extremely low attenuation at a frequency of 100 GHz and can be used successfully to transmit radiation over a distance of up to several meters.

## 3. Objects

Some light field distributions are complicated and require precise measurements. A measurement close behind the structure would be interesting in many cases, but the size of the packaging limits many detectors. To demonstrate the ability of the proposed method to register detailed distributions, we decided to examine the intensity pattern behind the 2D quasi-periodic structure forming its replica at particular distances (Talbot effect).

Two amplitude masks were designed and manufactured to verify the possibility of observing the Talbot effect at sub-THz frequencies. Both masks comprised a rectangular grid of square holes separated by the absorber. Masks were manufactured using FDM 3D printing technology from a highly absorptive composite material—carbon fiber-filled polyethylene terephthalate glycol (PETG)-based polymer blend. The period of these two grids (the distance between the centers of subsequent holes) was equal to 6 mm and 8 mm, respectively. The intensity maps and photographs of both masks manufactured are shown in Figure 5. Both objects were manufactured with a horizontal resolution equal to 400 μm, determined by the nozzle diameter (equal to 0.4 mm) used in the printing presses. The vertical resolution was equal to 100 μm, corresponding to the height of the print layer. During the printing process, the temperature of the nozzle was set to 250 ^∘^C, while the bed temperature was equal to 55 ^∘^C. The mentioned parameters allowed for good adhesion of the print, which was additionally enhanced by the brim application. The brim was removed from the object in the postprocessing of the print. The nozzle temperature of 250 ^∘^C allowed for the plasticization of the material and a relatively homogeneous distribution of carbon fiber (additive) throughout the print volume. It must be mentioned that incorrect printing parameters cause multiple unwanted effects that significantly decrease the print quality. The characteristic undesired effects for the PETG composite blends are nozzle clogging (due to the improper accumulation of carbon fiber particles in the nozzle), layer delamination in the print, and significant artifacts in the print, mostly the effect of the elevation of material along the nozzle during the printing. The material accumulated in this way adheres to the print, causing deformations.

## 4. Simulations

Due to the complexity of intensity patterns formed in the case of a Talbot effect, a set of simulations was carried out to enable a comparison with the experimental results. Simulations were performed for the design wavelength λ=95.3 GHz, with 0.25 mm sampling in the xy direction and 1 mm sampling in *z* direction (along the propagation axis) for two analyzed amplitude masks with periods equal to 6 mm and 8 mm. The propagation was calculated using a self-developed script written in Wolfram Language using the angular spectrum of plane waves method [37,38] and 1024 × 1024 pixel calculation matrices to determine the intensity distributions after propagation at a defined distance.

The self-imaging effect occurs under certain conditions [39], including the assumption of propagation in the near-field (Fresnel diffraction zone) and using particular objects obeying the Montgomery conditions [40] (all periodic patterns). In the case of THz radiation, one can encounter some differences in relation to visible light [9], and thus not all self-imaging conditions will always be fulfilled. Some issues should be discussed while describing the Talbot effect for THz frequencies, such as the following:The aperture size consists of a dozen wavelengths;The distance of propagation range is very close—ranging from single wavelengths to dozens of wavelengths;The amplitude object period size is in the range of single wavelengths;The object consists of a very limited amount of periods.

All these issues change the description of a well-known phenomenon for visible light when observing it in the THz range of frequencies. Thus, unique methods of detection and observation are needed to precisely determine the intensity pattern registered behind the object being self-imaged.

The first simulations were carried out for the amplitude objects with periods of d=6 mm and d=8 mm illuminated by a plane wave. The objects consisted of a limited number of periods, which does not significantly affect the quality of the self-imaging phenomenon [41]. Figure 6 and Figure 7 illustrate horizontal xz scans along the optical axis, as well as vertical xy scans perpendicular to the propagation axis simulated behind the object illuminated with a plane wave. It should be underlined that the xz scans must be shown for a particular *y* value, which means that they will vary between the levels at which they are visualized. The top panel in Figure 6 and the left and middle panels in Figure 7 show the xz intensity patterns plotted for two different values of *y*—one crossing the centers of the transparent openings of the mask and the second one shifted by $\frac{d}{2}$ and crossing the opaque parts of the mask. The bottom panel in Figure 6 and right panel in Figure 7 illustrate the xy scans—cross-sections perpendicular to the optical axis performed at particular *z* distances.

In the case of THz measurements, the illumination is usually a parallel or divergent beam from the source, mostly of the Gaussian type. Thus, the second simulations take into account that the structure is illuminated with a Gaussian beam originating 145 mm in front of the amplitude mask with the radius σ=60 mm at the position of the mask. The simulations are conducted according to plane wave simulations and consist of xz scans along the optical axis and xy scans that are perpendicular to the propagation axis illustrated in Figure 8 and Figure 9. The divergent wavefront is visible in the simulated intensity distributions in the form of an expanding intensity distribution with increasing propagation distance.

Talbot distance is defined as $z_{T}=\frac{2d^{2}}{\lambda }$ in the case of illumination of objects with a plane wave. Considering illumination with a divergent wavefront originating at a distance $z_{source}$ in front of the mask, the Talbot distance is scaled according to the following formula: $z_{T-divergent}=\frac{z_{source}\cdot z_{T}}{z_{source}-z_{T}}$. For the structure with a period equal to 6 mm and plane wave illumination, the quarter-, half-, and full-Talbot distances are equal to z=5.7 mm, 11.4 mm, and 22.9 mm, respectively, while in the case of divergent wave illumination, z=6.0 mm, 12.4 mm, and 27.2 mm, respectively. For the structure with a period equal to 8 mm and plane wave illumination, the quarter-, half-, and full-Talbot distances are equal to z=10.2 mm, 20.3 mm, and 40.7 mm, respectively, while in the case of divergent wave illumination, z=10.7 mm, 23.7 mm, and 56.6 mm, respectively. It should be underlined that the xy cross-sections in simulations are given for different distances than those resulting from divergent wave illumination calculations. The simulations were conducted for Gaussian-type beam illumination, which is slightly different from the divergent beam, but is more reliable for comparison with experimental evaluation. The simulations considered that the structure is illuminated with a Gaussian beam. It should be noticed that the divergence of the Gaussian beam is assumed in relation to the distance between the structure and the horn antenna, which the geometry of the horn might alter. Therefore, on the basis of conducted numerical simulations, the distances were adjusted to match the corresponding quarter-, half-, and full-Talbot images. These distances are 6 mm, 13 mm, and 28 mm for d=6 mm object and 10 mm, 18 mm, and 52 mm for d=8 mm object.

In the case of such complex diffraction patterns, additional simulations of intensity patterns are presented in Figure 10 for the xy planes corresponding to the experimentally registered values for two objects. As shown in Section 5, the experimental distances have been adjusted based on initial xz scans. The *z* axis values for scanning xy intensity distributions were determined according to the xz scans and their intensity distribution along the propagation axis *z*. As can be seen, these distances differ slightly from the theoretical ones. It should primarily be associated with the differences in the shape of the beam and the uncertainty of its position.

The differences between the theoretically calculated values and those obtained in the simulation and experiment may occur for several reasons. Some were already stated (illumination with the physical Gaussian-like beam instead of a theoretical divergent beam originating from a point source). Additionally, one should remember that the self-imaging effect is defined in a Fresnel diffraction zone, which is not fulfilled here. The requirement of small diffraction angles is not met in the described case.

In general, radiation propagation can be described through either the corpuscular theory of photons or the wave theory of electromagnetic radiation. In the case of a wave approach, a particular description is conducted that must take into account both propagation in free space and diffraction at obstacles. Depending on the propagation distance—*z*, the wavelength—λ, and the size of the obstacle (aperture), *D*, the following diffraction zones can be roughly distinguished:For $z>\frac{D^{2}}{4\lambda }$, the propagation is described by Fraunhofer integral—Fraunhofer zone (also called far-field);For $z\in (D,\frac{D^{2}}{4\lambda })$, the paraxial Fresnel integral is used—paraxial Fresnel zone (also called near-field);For z>>λ and z<D, the propagation is described by the Sommerfeld integral—non-paraxial zone;For z≈λ, the propagation is described by the Rayleigh–Sommerfeld integral—the zone very close to the aperture, also called Rayleigh zone.

The propagation distance ranges form some border between zones. The zones merge smoothly into one another and there is no sharp transition. Generally, two main regions with different behaviors can be distinguished—near-field and far-field.

In the case of our research, the aperture of the system is determined not by the object’s size (equal to 150 mm) but by the size of the illuminating beam. The Gaussian beam had particular divergence and the beam size was smaller than the object’s dimensions. We estimate it as a Gaussian beam with the radius σ=60 mm, so the Fresnel zone would start approximately 60 mm behind the structure. All our considerations are within the non-paraxial zone (z<D). It should be underlined that the convolution method used for calculating the propagation does not consist of any approximations, and using the transfer function is not limited to the paraxial case—it allows us to obtain proper results in the non-paraxial zone, as in case of the described experiment. On the other hand, the very close zone, where z≈λ, is outside the applicability range for this method.

## 5. Experiment

The experimental setup consisted of an IMPATT diode-based source from Terasense Group, Inc., San Jose, CA, USA, emitting the radiation at 95.3 GHz with an optical power of 900 mW. The source was equipped with a WR-10 horn antenna (25 dB of gain, +/− 7 degrees directivity, according to the manufacturer). The IMPATT source emits strongly coherent linearly polarized radiation [42]. This results in parasitic interferences occurring between the signal and the beams reflected from any surface in the system. Therefore, any experimental setup must be carefully configured to avoid unnecessary perpendicular surfaces, which also shows the advantage of the proposed waveguide scanning method. The amplitude masks have been placed at a distance of 145 mm from the tip of the antenna. In this way, the divergent wavefront originating from the source illuminated the designed masks. The optical field detection behind the masks was realized using two methods. The first served as a reference and used the Terasense Tera-1024 camera. The camera gathers information on the intensity of the terahertz radiation in a matrix of 32 × 32 pixels with a pitch of 1.5 mm. The second method, investigated in this paper, revolves around applying polymer filaments as waveguides of the THz radiation. The photograph of this setup is shown in Figure 11.

The 1.75 mm thick tip of the COC filament was placed directly in front of the mask to probe the optical field. The polymer waveguide was bent to redirect the coupled radiation from the path of the illuminating radiation. The other end of the waveguide was placed in front of the VDI PM4 power meter. A horn antenna equipped with a dedicated 3D-printed waveguide coupler was used to deliver the signal to the detector.

## 6. Results

The two scanning methods described above were deployed for both the structures we analyzed. In all four experimental cases, the horizontal xz scan was performed first to gather information about the evolution of the optical field behind the structures. Next, perpendicular xy scans were completed in the planes corresponding to the predicted positions of the quarter-, half-, and full-Talbot distances.

The scans performed with the camera behind the d=6 mm structure are shown in Figure 12.

The horizontal xz scan performed in the plane intersecting the centers of the openings in the evaluated structure is shown on the left. The perpendicular xy scans are shown on the right and correspond to the top to bottom to the full, half, and quarter of the Talbot distances. The distances at which these scans were obtained are also marked with black dashed lines in the xz scan. The presence of distorted self-images of the mask can be observed in the form of radially arranged sections at distances between 20 mm and 50 mm from the structure. The perpendicular scans performed at z=21 mm and z=12 mm clearly show the presence of direct and phase-shifted self-images of the mask. The presence of the sub-image at the quarter of the Talbot distance (z=6 mm) is debatable.

Analogous scans were also performed behind the d=8 mm structure and are shown in Figure 13. The panels here are arranged in the same way as in Figure 12. The only change is the estimation of the distances for particular self-images, which depends on the constant of the mask. The white lines visible near the top of the xz scan arise from camera malfunction and do not influence the obtained results. In this case, the spacing between the bright spots is proportionally higher (8 mm vs. 6 mm), which theoretically facilitates their detection with a fixed-resolution camera. However, this comes at the cost of illuminating a smaller number of periods of the masks, which impairs the Talbot effect. All in all, once again, the effect is observable, although the details of the field distribution are hard to determine.

This was the starting point for developing the waveguide scanning method, allowing for a much more dense subwavelength probing of the terahertz field. Analogous scans performed with the waveguide scanning method behind the d=6 mm structure are shown in Figure 14.

As can be seen, the improvement in the resolution of the scans is substantial. The details of the Talbot-like carpet are much more fine. In addition, the self-images in the xy scans are much more apparent. The proposed scanning method allowed one to register the intensity pattern at very close distances behind the periodic mask. It is crucial because a very near field can be observed, which will help to develop a proper theoretical description—discussing the self-imaging properties in the zone where z≈λ. The simulation results show the accordance with the typical Talbot effect for half- and full-Talbot distances, while for the quarter-Talbot distance, the uniformity and proper scaling of the pattern is not observed for all cases. The experimental evaluation with higher resolution allowed us to observe the shift by half of the period in the self-images for half- and full-Talbot distances—xy intensity patterns illustrate this effect. Still, predominantly in xz scans, the switching between different places of the self-imaged mask is visible.

Structures with larger periods form the self-images at further distances, so switching self-images for half- and full-Talbot distances is clearly visible.

A very interesting fact is the lack of a typical distribution for a Talbot effect complicated by the light field distribution behind the mask, corresponding to the so-called fractional Talbot effect. In Figure 14, the xy scan corresponding to z=6 mm illustrates the intensity pattern characteristic to the half-Talbot distance distribution. We attribute such behavior to the fact that the mask period has a size of around 2 wavelengths, and the observation is carried out at the *z* distance, also equal to 2 wavelengths. This phenomenon is less visible for structures with larger periods (d=8 mm) illustrated in Figure 15, which is consistent with our assumptions. Here, the intensity pattern of xz reveals changes in the fractional distances of the Talbot effect.

The same procedure has also been performed for the d=8 mm structure, shown in Figure 15. Also, in this case, the improved scanning resolution allowed for the registration of the self-image and the shifted self-image of the mask.

The improvement in subwavelength imaging can be noticed directly in the intensity cross-sections shown in Figure 16.

The data obtained for the d=6 mm mask are presented—cross-sections along the dotted lines in Figure 12 and Figure 14. The data from both scanning methods were normalized for clear comparison. For closer distances (Figure 16a,b), one can notice the overexposure of the camera, which is unavoidable in the case of a low-dynamic range detector and high-dynamic range fields. It shows an additional advantage of the waveguide scanning method, which can be used with various detectors and tailored for specific use. Most importantly, however, all three scans show a significant improvement in the resolution when using the waveguide scanning method, proving the exceedance of the subwavelength scale. The wavelength is given for scale (red arrows in Figure 16) for easier comparison.

## 7. Conclusions

The waveguide scanning method has a significantly higher threshold of detection. It is embedded in the concept of the detection scheme, which aims at a maximal increase in resolution at the cost of other detection parameters. However, as in the investigated case, the method shines when sufficient optical power is delivered. The improvement in contrast and ability to resolve and distinguish details of the obtained scans is tremendous.

The Talbot effect has been observed for sub-THz frequencies. Compared to the higher frequencies, in the case of the sub-THz band, the self-images are separated by a relatively high distance (in order of several millimeters), which could allow for separate manipulation of particular spots. Using the exact and shifted self-images switches the beam between two spatially separated distributions. It opens up many application opportunities, e.g., in filtering or radiation multiplexing.

The waveguide scanning method allowed for the registration of a complex radiation pattern formed behind a quasi-periodic mask along the propagation axis, also enabling scanning in close proximity to the mask. It should be underlined that the aperture size of a mask consists only of a dozen wavelengths, the mask period size is in the range of only single wavelengths, and it consists of a very limited number of periods. Moreover, the propagation distance is very close, ranging from single wavelengths to dozens of wavelengths. These issues make the observation of the self-imaging effect in the sub-THz range demanding.

The waveguide scanning method for the detection of terahertz radiation has been demonstrated using straightforward and cost-effective means. It significantly improves the scanning resolution (reaching subwavelength resolution). Moreover, this method can be used when an approach with a bulky detector is impossible in the investigated area. It also allows for the detection of radiation very close to the examined structures, which may include narrow spaces or volumes filled with liquids or hazardous gases. Another application idea is to mitigate heat transfer in the bolometric detectors [43]. The limitation of the proposed method lies mainly in the guiding and coupling efficiency of the radiation. As the transmission losses in COC are marginal (in the sub-THz range), distances up to several meters can be achieved. The problem lies in a very sensitive guiding (a significant part of the mode is guided outside of the filament). In order to obtain the presented results, self-developed holders were utilized, which were 3D-printed out of PP or made from various foams. Catching the filament with, for example, a metallic holder would result in significant losses.

## Figures and Tables

**Figure 1 sensors-25-00336-f001:**
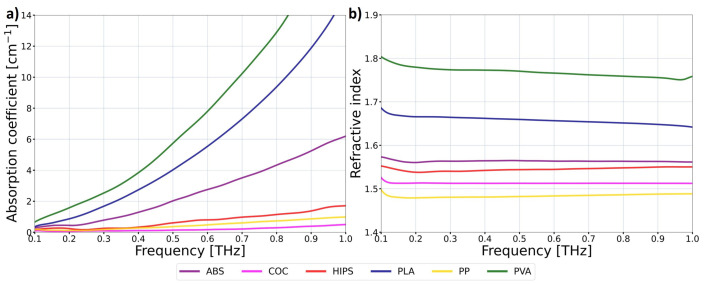
The optical properties of various polymer materials in THz frequency domain measured with THz time-domain spectroscopy (TDS). The presented data concerns (**a**) the absorption coefficient of the materials and (**b**) the refractive index of the materials.

**Figure 2 sensors-25-00336-f002:**
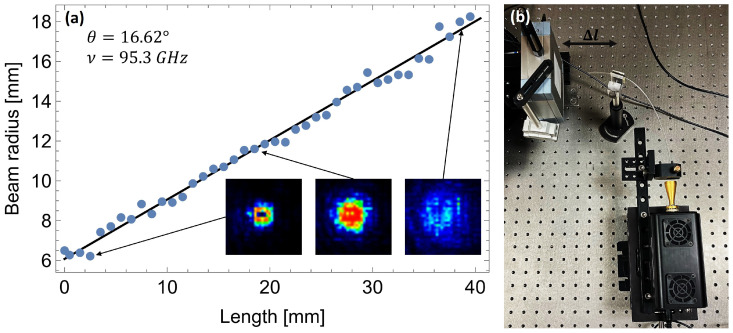
(**a**) The dependence of the beam’s radius leaving the waveguide on the distance from its end. The fitted line shows the half-angle of the emitted cone of radiation. In the inset, the beam images obtained at three different distances are shown. (**b**) The photograph of the setup used in the experiment.

**Figure 3 sensors-25-00336-f003:**
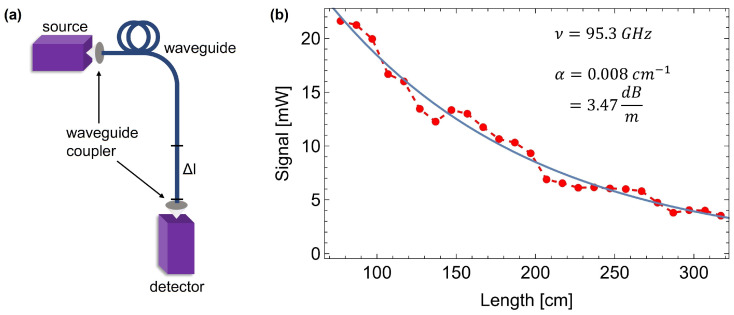
(**a**) The scheme of the setup used in the experiment to measure attenuation. (**b**) The dependence of the radiation power measured at the end of the waveguide section on its length (red markers). The fitted exponential curve (blue line) defines the attenuation of the waveguide.

**Figure 4 sensors-25-00336-f004:**
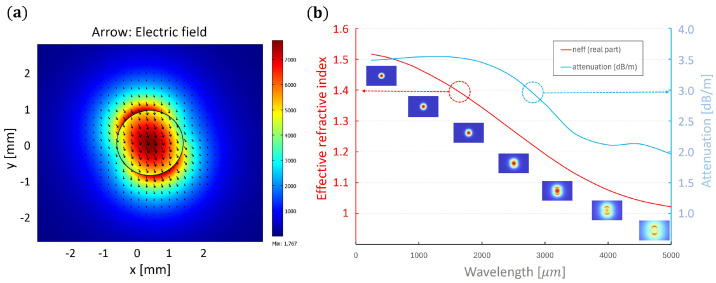
(**a**) Distribution of the normalized electric field of a mode guided within the waveguide and (**b**) the dependence of the effective refractive index (red) and attenuation coefficient (blue) on the frequency of the guided radiation with mode profiles corresponding to given frequencies in the insets.

**Figure 5 sensors-25-00336-f005:**
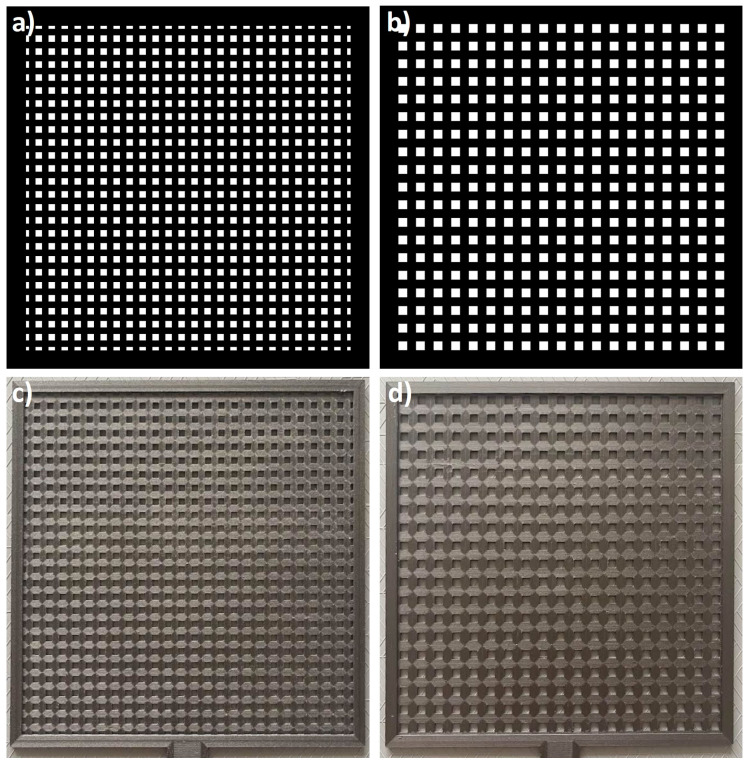
(**a**,**b**) Grayscale images representing binary masks of periodic structures in the 6 mm and 8 mm period; (**c**,**d**) manufactured binary masks using FDM 3D-printing technology from the highly absorptive composite of PETG with carbon fiber additive.

**Figure 6 sensors-25-00336-f006:**
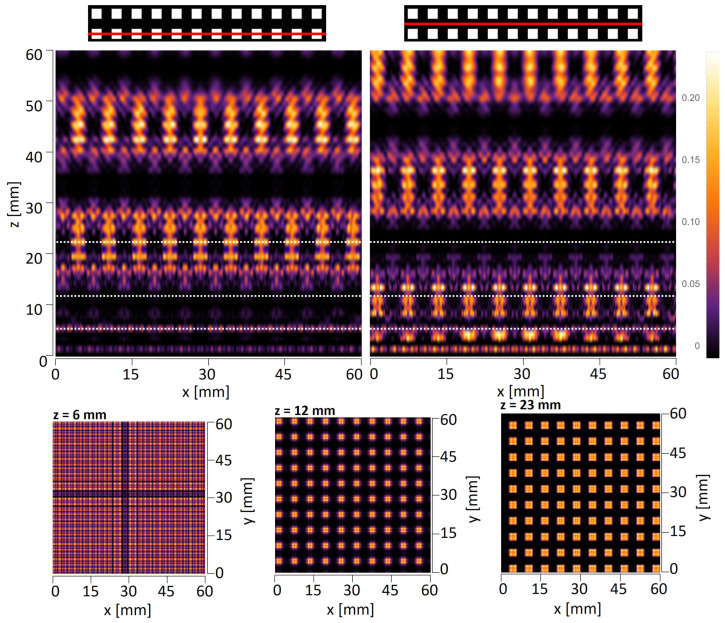
The simulated intensity distributions observed behind the quasi-periodic structure having a 6 mm period illuminated with a plane wave. The top panel illustrates xz distributions plotted for two different *y* values—left crossing next to openings of the object and right crossing the region without openings. White dotted lines indicate the distance of xy cross-sections. The bottom panel illustrates xy intensity distributions at three distances corresponding to the quarter-, half-, and full-Talbot distances.

**Figure 7 sensors-25-00336-f007:**
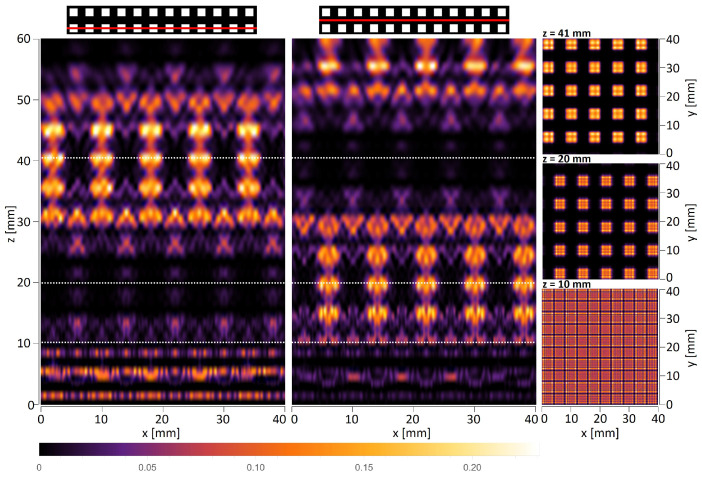
The simulated intensity distributions observed behind the quasi-periodic structure having an 8 mm period illuminated with a plane wave. The plots on the left illustrate xz distributions for two different *y* values—left crossing next to openings of the object and right crossing the region without openings. White dotted lines indicate the distance of xy cross-sections. The plots on the right illustrate xy intensity distributions at three different distances corresponding to the quarter-, half-, and full-Talbot distances.

**Figure 8 sensors-25-00336-f008:**
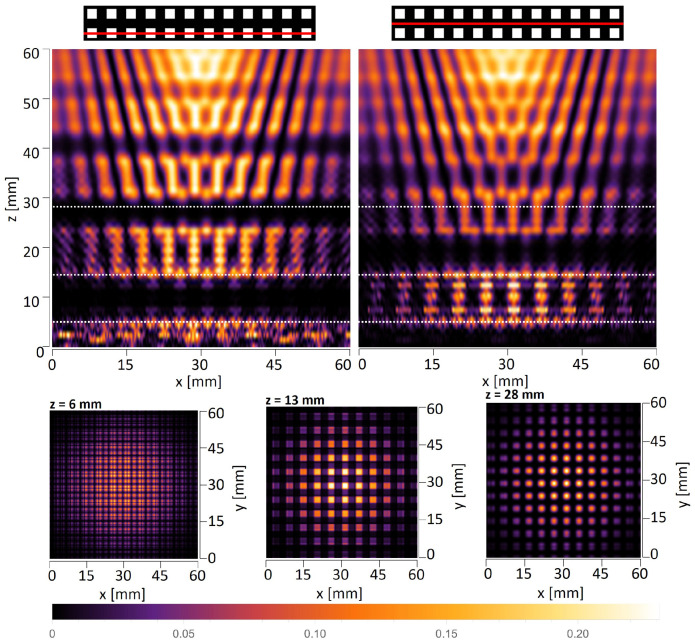
The simulated intensity distributions observed behind the quasi-periodic structure having 6 mm period illuminated with Gaussian type beam originating 145 mm in front of the object and having the radius σ=60 mm at the position of the object. The top panel illustrates xz distributions plotted for two different *y* values—left crossing next to openings of the object and right crossing the region without openings. White dotted lines indicate the distance of xy cross-sections. The bottom panel illustrates xy intensity distributions at three different distances corresponding to the quarter-, half-, and full-Talbot distances.

**Figure 9 sensors-25-00336-f009:**
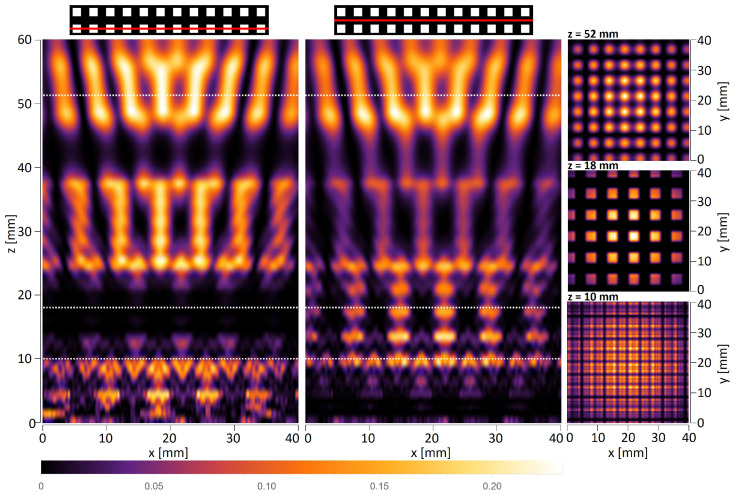
The simulated intensity distributions observed behind the quasi-periodic structure having 8 mm period illuminated with Gaussian type beam originating 145 mm in front of the object and having the radius σ=60 mm at the position of the object. The plots on the left illustrate xz distributions for two different *y* values—left crossing next to openings of the object and right crossing the region without openings. White dotted lines indicate the distance of xy cross-sections. The plots on the right illustrate xy intensity distributions at three different distances corresponding to the quarter-, half-, and full-Talbot distances.

**Figure 10 sensors-25-00336-f010:**
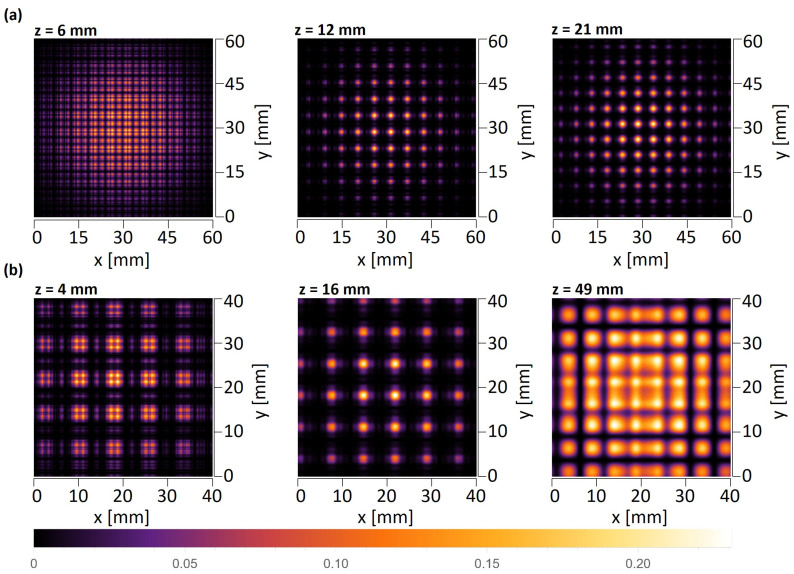
The xy intensity distributions for two objects (periods d=6 mm and d=8 mm), simulated at the distances corresponding to quarter-, half-, and full-Talbot distances and taking into account different illumination in the experimental evaluation.

**Figure 11 sensors-25-00336-f011:**
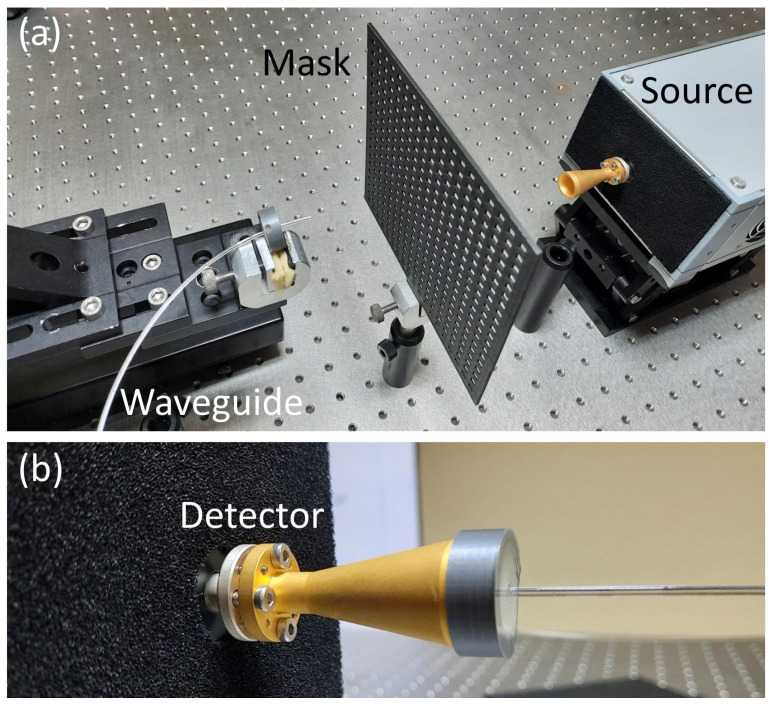
The photographs of the experimental setup. Panel (**a**) shows the mask illuminated with the divergent wavefront emitted from the IMPATT source and the waveguide tip placed on the motorized 3D stage used for scanning. Panel (**b**) shows the other end of the waveguide placed in the dedicated 3D-printed holder attached to the detector’s horn antenna.

**Figure 12 sensors-25-00336-f012:**
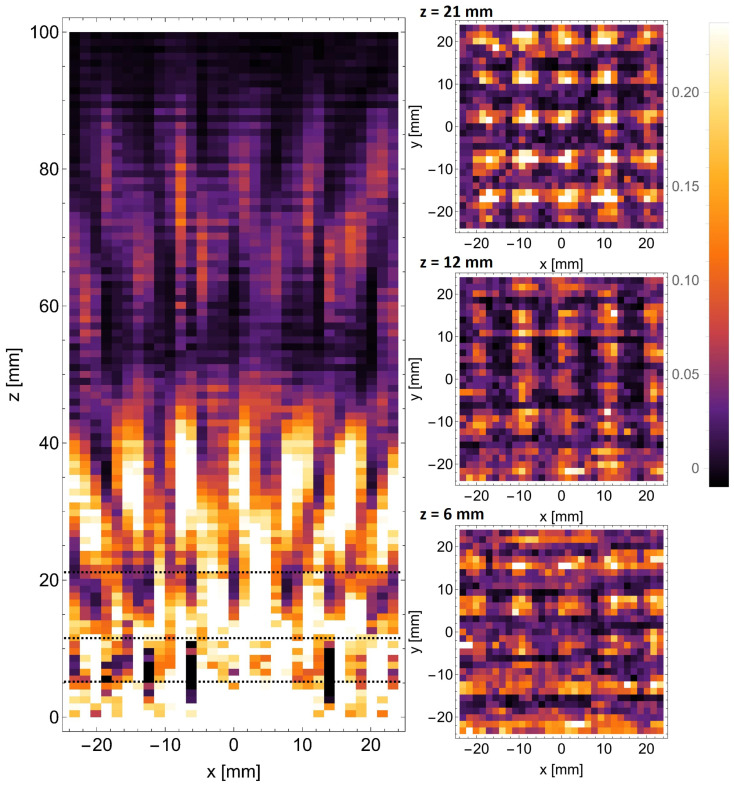
The scans performed with THz camera behind the d=6 mm mask. The horizontal scan is shown on the left, accompanied by the vertical scans performed in the marked positions.

**Figure 13 sensors-25-00336-f013:**
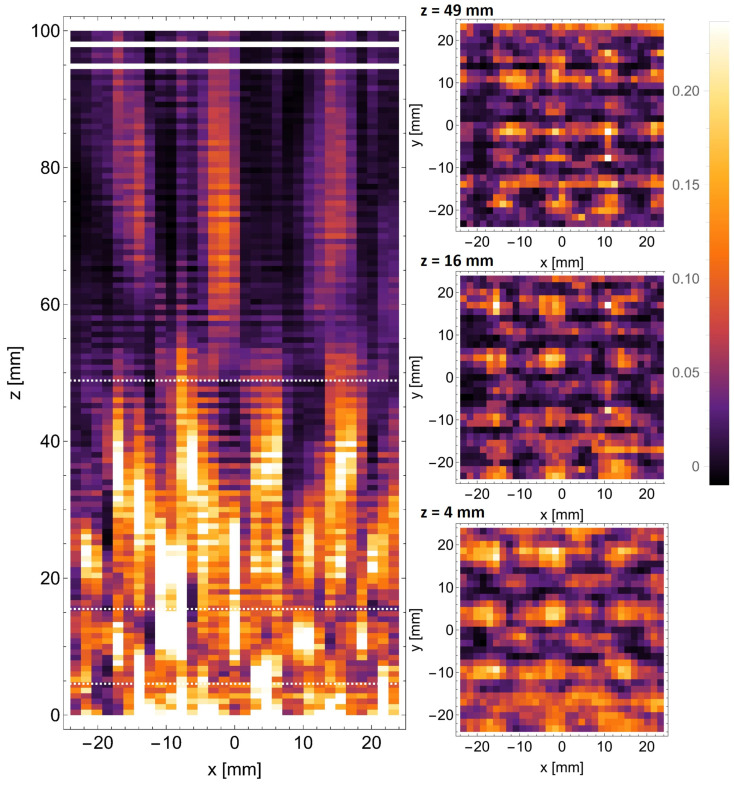
The scans performed with THz camera behind the d=8 mm mask. The horizontal scan is shown on the left, accompanied by the vertical scans performed in the marked positions.

**Figure 14 sensors-25-00336-f014:**
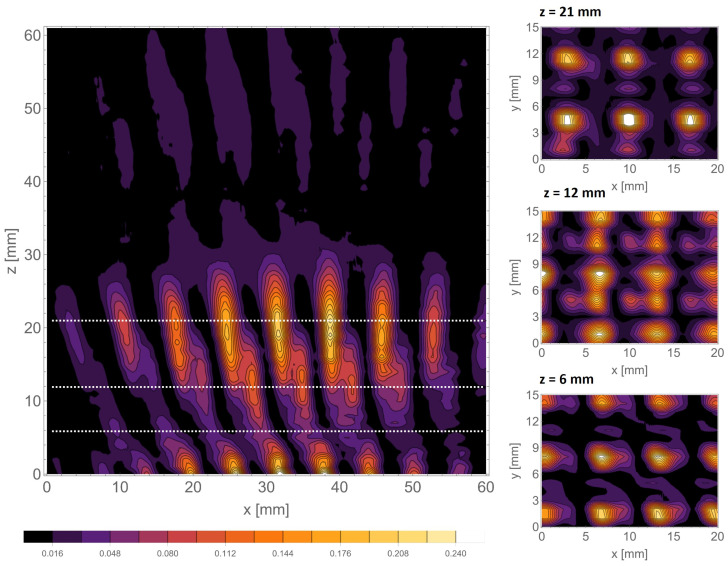
The scans performed with the waveguide scanning method behind the d=6 mm mask. The horizontal scan is shown on the left, accompanied by the vertical scans performed in the marked positions.

**Figure 15 sensors-25-00336-f015:**
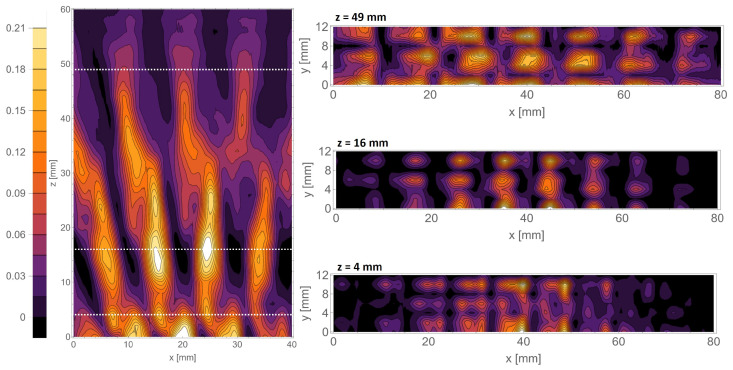
The scans performed with the waveguide scanning method behind the d=8 mm mask. The horizontal scan is shown on the left, accompanied by the vertical scans performed in the marked positions.

**Figure 16 sensors-25-00336-f016:**
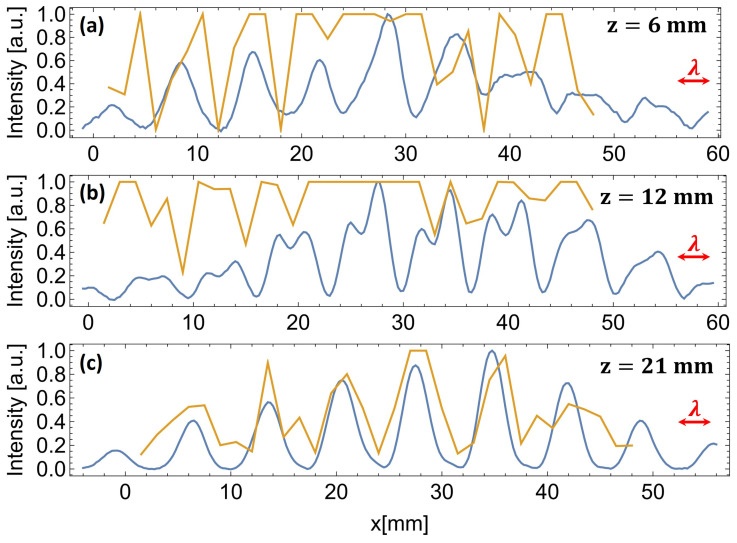
The cross-sections of the registered intensity patterns obtained for the d=6 mm mask at distances of (**a**) z=6 mm, (**b**) z=12 mm, and (**c**) z=21 mm from the mask. The wavelength (λ) is included for scale as a red arrow. The blue lines indicate the waveguide scanning system, and the yellow lines indicate the camera.

## Data Availability

The raw data supporting the conclusions of this article will be made available by the authors on request.

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
