# Peer review of "Subwavelength Imaging in Sub-THz Range Using Dielectric Waveguide"

_sensors, 2025, doi:10.3390/s25020336_

Round 1
Reviewer 1 Report
Comments and Suggestions for Authors
The authors use waveguides to realize the scanning imaging characteristics of terahertz waves.
1. The author can describe the waveguide connector in more detail, that is, the link between the waveguide and the horn antenna.
2. Why is the simulated field area different from the experimentally measured field area?
3. The back view of the beam control device should be increased. The research background given by the author is a little insufficient. For example: Opto-Electron Adv 6, 220012 (2023).Opto-Electron Sci 2, 220026 (2023). Opto-Electron Adv 6, 220073 (2023).Opto-Electron Sci 2, 220019 (2023).
Author Response
Response in the pdf file.

Reviewer 2 Report
Comments and Suggestions for Authors
This work presents a comprehensive study, which includes a detailed examination of a waveguide scanning method and the sub-THz Talbot effect. The paper opens with an extensive introduction, in which the necessity for waveguide at the THz range is elucidated, the current deficiencies of THz cameras are outlined, and a novel method for waveguide scanning is put forward. The principal findings are presented in five sections. The initial section of the study examines the various materials that could be employed in the fabrication of a waveguide, with a particular focus on the properties of cyclic olefin copolymer waveguides, which exhibited superior transmission characteristics. The subsequent section detailed the creation of masks for observation of the Talbot effect. The third section introduces the theory and simulations of the Talbot effect, demonstrating the transmission of plane and Gaussian waves through the selected masks. The fourth section was dedicated to the presentation of the experimental setup. The final section, entitled 'Results', presents the intensity profiles obtained following the transmittance of the masks through the waveguide, as observed by the THz camera and with the use of a detector with the waveguide. The results chapter concludes with a comparison between the waveguide scanning method and a commercial THz camera, which demonstrates that the former exhibits superior resolution. Comprehensive conclusions are represented in the end of the paper. Overall, the work is a solid contribution to the field, demonstrating the construction of a functional sub-THz waveguide made of cyclic olefin copolymer and illustrating the potential for significant resolution enhancement through the use of the waveguide scanning method.
The English is clear and the style is of an acceptable standard. However, there is a minor issue in line 32, where the abbreviation FMCW is not expanded in either the abbreviation section or the surrounding text.
I have some minor comments:
1. Is your IMPATT detector a coherent source? If so it could be elaborated more in the text.
2.In fig. 16 the blue and yellow lines are not mentioned in the description of the picture. Although its implied in the text that yellow is for THz camera and blue for waveguide detection.
Author Response
Response in the pdf file.

Reviewer 3 Report
Comments and Suggestions for Authors
The dielectric waveguide for scanning resolution is good to be observed. Therefore, the authors described waveguide scanning method in THz frequency ranges. The related work for THz waveguides is well summarized. Figure quality looks fine. Simulated and experimental results performed with THz camera looks reasonable. Overall, the manuscript is well written. Therefore, I can recommend the submitted manuscript is minor revision with suggestive comments.
1. The authors showed cross-sections of the registered intensity patterns in Figure 16 with different z. How about the results over 21 mm ?
2. Figure 4 fonts seem to be small to be observed. Please enlarge that.
3. Please use Figure instead of Fig. Please see authors' guidelines of the MDPI.
4. The authors had better compare the important or meaningful data with other scholars' work in a Table.
5. What is the technology limitation of the proposed waveguide scanning method ?
6. The abbreviated journal names in reference section should be used.
7. In abbreviations section, there are same 4 PVA definition. Please delete them.
8. In Figure 11, the authors showed some experimental setup. However, the authors do not describe the information of the components such as company information and specifications of the components.
9. In Figure 5, the authors showed some masks of the periodic structures. As far as I know, some materials in mask are important to be selected due to absoprtion. How to determine those strucures in the mask ?
10. In Figure 5, the authors selected the retangular patterns. If the patterns are not retangular patterns, how the authors expect the the issues of the radition pattern for scanning resolution ?
Author Response
Response in the pdf file.

Round 2
Reviewer 1 Report
Comments and Suggestions for Authors
It can be accepted.